# *Lactiplantibacillus plantarum* GOLDGUT-HNU082 Alleviates CUMS-Induced Depressive-like Behaviors in Mice by Modulating the Gut Microbiota and Neurotransmitter Levels

**DOI:** 10.3390/foods14050813

**Published:** 2025-02-26

**Authors:** Wanggao Li, Meng Xu, Yaning Liu, Silu Zhang, Jun Wang, Zhizhu Zhang, Guoxun Xiao, Ruimin Wang, Jiachao Zhang, Hui Xue

**Affiliations:** 1Key Laboratory of Food Nutrition and Functional Food of Hainan Province, School of Food Science and Engineering, Hainan University, Haikou 570228, China; lwg15277991592@163.com (W.L.); xemeng950217@163.com (M.X.); 13574035298@163.com (Y.L.); wangruimin2022@163.com (R.W.); jiachao@hainanu.edu.cn (J.Z.); 2Collaborative Innovation Center of One Health, Hainan University, Haikou 570228, China; 3Wonderlab Innovation Centre for Healthcare, Shenzhen Porshealth Bioengineering Co., Ltd., Shenzhen 518000, China; echo.zhang@wonderlab.top (S.Z.); junwang@wonderlab.top (J.W.); zhizhu.zhang@wonderlab.top (Z.Z.); shawn@wonderlab.top (G.X.)

**Keywords:** depression, functional food, *Lactiplantibacillus plantarum* GOLDGUT-HNU082, gut–brain axis, serotonin

## Abstract

Emerging evidence links depressive disorders to the gut microbiota via the gut–brain axis. Probiotics, which are microorganisms that modulate the gut microbiota, have shown promising results in alleviating depression and are increasingly recognized as functional food components with potential health benefits. This study examines the effects of *Lactiplantibacillus plantarum* GOLDGUT-HNU082 (Lp082), a probiotic strain with potential applications in functional foods, on chronic unpredictable mild stress (CUMS)-induced depression in mice. Behavioral tests, measurements of the neurotransmitters and inflammatory cytokines in the serum and colon tissue, and the metagenomic sequencing of the gut microbiota were used to investigate potential mechanisms. The results demonstrated that Lp082 significantly alleviated depressive-like behaviors in CUMS mice, restored the balance of key neurotransmitters like serotonin (5-HT), reduced the levels of inflammatory cytokines like TNF-α, and enhanced brain neuroplasticity by promoting hippocampal neurogenesis. Additionally, Lp082 altered the composition of the gut microbiota in CUMS mice and promoted the growth of *Bifidobacterium*, improving metabolic pathways related to neurotransmitter synthesis. These findings indicate that Lp082, as a potential functional food ingredient, alleviates depressive-like behaviors in mice by reshaping the gut microbiota, offering new insights into the use of probiotics in functional foods for mental health management.

## 1. Introduction

Depression has become increasingly common worldwide in recent years, affecting more than 280 million people [1]. In mild cases, it interferes with daily activities and work. In severe cases, it can result in self-harm or even suicide. Self-harming behaviors in individuals with depression have increasingly become a major contributor to disability, warranting close attention [2]. While existing treatments, such as psychotherapy and pharmacotherapy, have alleviated symptoms to some extent, they still face significant limitations, including limited efficacy and notable side effects. Antidepressants with relatively high efficacy, such as agomelatine, escitalopram, vortioxetine, and fluoxetine, are still associated with high treatment failure rates and side effects, including nausea, rashes, and liver or kidney damage [3]. Therefore, the exploration of effective strategies for the treatment of depression is critically important.

A growing number of studies have demonstrated the significant role of the gut–brain axis (GBA) in regulating mood and behavior. The GBA refers to the interactions between the gut and the central nervous system (CNS) [4]. The gut microbiota plays an important role in mood regulation through its effects on the nervous, immune, endocrine, and metabolic systems. The gut microbiota can produce or regulate neurotransmitters and metabolites, such as serotonin (5-HT), which influence mood and behavior through the enteric nervous system [5]. Therefore, the regulation of mood via the GBA has emerged as a potential strategy for the development of antidepressant therapies.

Probiotics, which modulate the gut microbiota, have shown promising antidepressant effects. A growing number of studies have shown that probiotics, such as *Lactobacillus helveticus*, *Lactobacillus rhamnosus*, and *Bifidobacterium longum*, can improve depressive-like behaviors in patients [6,7,8]. In a clinical trial, supplementation with *B. longum* CCFM1025 significantly alleviated depression by altering the tryptophan metabolic pathway [9]. In another clinical study, supplementation with probiotic Lactobacillus plantarum 299v improved metabolic disorders and increased the levels of oxidized glycerophosphocholine (oxPC) in patients with major depression [10]. The chronic unpredictable mild stress (CUMS) protocol is a commonly used animal model to induce depression in mice by exposing them to a series of random, mild stressors encountered in daily life. Research by Chevalier and colleagues demonstrated that CUMS-induced depressive mice showed a significant reduction in the *Lactobacillus* abundance in their gut microbiotas. After fecal microbiota transplantation, the depressive-like behavior of the mice improved significantly, accompanied by a substantial increase in the *Lactobacillus* abundance, suggesting that the gut microbiota influences depressive phenotypes through the modulation of the endocannabinoid system [1]. Additionally, research by Zhao and colleagues showed that supplementation with *Lactiplantibacillus plantarum* R6-3 could regulate the gut microbiota, improve the immune and oxidative stress responses, and promote the production of monoamine neurotransmitters, thereby alleviating depressive-like behaviors in mice [11].

*L. plantarum* GOLDGUT-HNU082 (Lp082), isolated from fermented Yucha in Hainan Province, China, is acid- and bile salt-tolerant [12]. Whole-genome sequencing has suggested that Lp082 holds great potential as a probiotic with promising physiological functions [13]. The results obtained by Huang et al. showed that Lp082 enhances the gut microbiota’s ecological and genetic stability [14], inhibits *Fusobacterium’s* growth, reduces inflammation [15], and alleviates ulcerative colitis through gut microbiota modulation [16]. Furthermore, Lp082 has been shown to regulate metabolic pathways to prevent hyperlipidemia [17]. Lp082 can also modulate the gut microbiota in zebrafish and promote the synthesis of 5-HT, indicating its potential antidepressant properties [18].

This study investigates the potential of Lp082 as a functional food ingredient to mitigate depression-associated pathophysiology by modulating the gut–brain axis’ signaling and neuroinflammatory pathways. To evaluate its dietary intervention potential, behavioral tests were conducted on CUMS model mice, complemented by quantitative analyses of serum/brain neurotransmitters (5-HT, GABA) and inflammatory markers (TNF-α, IL-6). Integrated gut microbiota profiling (shotgun metagenomics) and metabolomics analyses were employed to systematically assess the effects of Lp082 on the host physiology and diet–microbe–host interactions. These findings may advance the development of microbiota-targeted functional foods for mental health management.

## 2. Materials and Methods

### 2.1. Materials and Animals

In a prior study, Lp082 was isolated from fermented Yucha in Hainan Province, China, and stored at −80 °C. Yucha was homogenized with saline at 1:9 to form a suspension, followed by gradient dilution. Subsequently, 100 μL of each suspension with different gradient concentrations was coated on MRS (Guangdong Huankai Microbial Technology Co., Ltd., Guangzhou, China) AGAR medium and cultured at 37 °C for 2 days to isolate Lactobacillus [12].

Twenty-four male C57BL/6J mice, aged 5 weeks, were obtained from the Beijing Vital River Laboratory Animal Technology Co., Ltd. (Beijing, China). The standard diet, containing 22.9% protein, 11.1% fat, and 66% carbohydrates, was provided by the Jiangsu Xietong Pharmaceutical Biotechnology Co., Ltd. (Nanjing, China).

### 2.2. Experimental Design

The animal study was approved by the Ethics Committee of Hainan University (approval number HNUAUCC-2023-00183, 5 September 2023). Mice were housed in specific pathogen-free (SPF)-grade animal facilities under a 12 h light/dark cycle at 23 ± 2 °C and 55 ± 5% humidity. After 1 week of acclimatization, the mice were randomly divided into three groups: the Con group (*n =* 8), Mod group (*n =* 8), and Lp082 group (*n =* 8). Except for the Con group, the mice were exposed to unpredictable stressors daily for 8 weeks. The stressors included wet bedding, no bedding, restraints, cage tilting, light/dark cycle alterations, flashing lights, and a 4 °C cold water bath. On average, two stressors were applied each day; a detailed schedule is provided in Appendix A. During this period, the Con and Mod groups were administered 200 μL of physiological saline via gavage daily at 10:00 AM, while the Lp082 group received 200 μL of an Lp082 suspension (1 × 10^9^ CFU) via the same method daily at 10:00 AM. After 8 weeks, behavioral tests were performed.

### 2.3. Animal Behavioral Tests

#### 2.3.1. Open Field Test (OFT)

The OFT was conducted in a well-insulated laboratory with a controlled temperature, humidity, and light intensity. After 1 h of adaptation, the mouse was gently placed in the bottom of the open field box. The activity of the mice was recorded by a video camera for the following 8 min [19]. The software VisuTrack v.3 (Shanghai Xinruan Information Technology Co., Ltd., Shanghai, China) was utilized to analyze the recorded videos of the animal behavior experiments. Before each open field test, the box was disinfected with 75% ethanol to eliminate odors.

#### 2.3.2. Elevated Plus Maze Test (EPM)

The mouse was placed in the maze, ensuring that each mouse was positioned in the same location throughout the experiment, and their activity was recorded by a video camera for the following 5 min. The open arm time was analyzed using the VisuTrack software. During the experiment, the experimenter maintained a distance of 1 m from the maze.

#### 2.3.3. Sucrose Preference Test (SPT)

The SPT consisted of two parts, an adaptation training part and a test part. One bottle of distilled water and one bottle of 1% (*w*/*v*) sucrose solution were placed in each cage during the 48 h adaptation training phase, during which the position of the water bottle was switched. After the adaptive training, the mice were fasted and water-deprived for 24 h. Subsequently, the mice were given distilled water and sucrose aqueous solutions for a 12 h test, and their sucrose solution consumption and distilled water consumption were recorded after 12 h.

#### 2.3.4. Tail Suspension Test (TST)

During the tail suspension test, the animals were subjected to a 6-min trial protocol, with the initial 2 min designated as a habituation period and the subsequent immobility duration quantified over the final 4-min observational interval [20]. A longer immobility time indicates higher levels of depression, as the mouse attempts to escape less frequently during the latter part of the test.

#### 2.3.5. Forced Swim Test (FST)

The mice were placed individually into a plastic cylinder filled with water to a depth of 18 cm at 23–25 °C for 5 min. Their activity was recorded using a camera, and the software was used to automatically assess the immobility time. A longer immobility time indicates higher levels of depression.

### 2.4. Animal Sample Collection

Following the behavioral tests, fresh fecal samples were aseptically collected for shotgun metagenomic sequencing. After a 12 h fasting period, blood samples were obtained via retro-orbital bleeding under anesthesia, followed by euthanasia through cervical dislocation. Whole blood was allowed to clot at room temperature for 2 h prior to centrifugation (3000 rpm, 20 min, 4 °C) for serum isolation, which was aliquoted and archived at −80 °C until biochemical analyses. Brain and colonic tissue was immediately excised and divided into two aliquots: one fixed in 4% paraformaldehyde for histopathological examination and the other snap-frozen in liquid nitrogen for molecular studies.

### 2.5. Physiological and Biochemical Assessments

The levels of 5-HT, tumor necrosis factor-alpha (TNF-α), dopamine (DA), interleukin (IL)-6, acetylcholine (Ach), gamma-aminobutyric acid (GABA), glutamate (Glu), NE, IL-1β, and lipopolysaccharide (LPS) in the serum, colon, and brain tissue were measured using ELISA kits from the Shanghai Xinyu Biotech Co., Ltd. (Shanghai, China).

### 2.6. Histopathological Analysis

Hippocampal tissue was fixed in 4% paraformaldehyde, dehydrated, embedded in paraffin, and sectioned. Nissl staining was performed to observe the neuronal cellular structures. After deparaffinization, antigen retrieval, and the blocking of nonspecific binding, the sections were incubated overnight with primary antibodies against brain-derived neurotrophic factor (BDNF), doublecortin (DCX), synaptophysin, postsynaptic density protein 95 (PSD-95), and glial fibrillary acidic protein (GFAP), conjugated with red or green fluorescent labels. The sections were then incubated with fluorescently labeled secondary antibodies, stained with DAPI to visualize the nuclei, and mounted with antifade reagent for fluorescence microscopy. The antibodies and reagents used for the histopathological analysis were provided by the Wuhan Servicebio Technology Co., Ltd. (Wuhan, China).

### 2.7. Fecal Metagenomic Sequencing and Microbial Species Annotation

Total DNA was extracted from the mouse fecal samples using the QIAamp DNA Stool Mini Kit (Qiagen, Hilden, Germany) according to the manufacturer’s instructions. The DNA’s purity and integrity were assessed. Sequencing libraries were prepared using the Illumina NEBNext^®^ Ultra DNA Library Prep Kit (NEB, Ipswich, MA, USA), and DNA fragments of approximately 350 bp were obtained via ultrasonication. Shotgun metagenomic sequencing was conducted on the Illumina NovaSeq 2500 platform at the Novogene Co., Ltd. (Beijing, China). The metagenomic data were analyzed using the HUMAnN3 v.3 software, and microbial species were identified. Metabolic pathways and abundance information were predicted based on the MetaCyc database.

### 2.8. Serum Metabolomics

The experimental samples were first thawed at 4 °C, followed by vortexing for 1 min to ensure thorough mixing. A suitable volume of the sample was transferred to a 2 mL centrifuge tube, to which 400 µL of methanol solution was added. After vortexing for another minute, the sample was centrifuged at 12,000 rpm for 10 min at 4 °C. The supernatant was carefully decanted into a new 2 mL centrifuge tube and concentrated to dryness. The residue was then re-dissolved in 150 µL of an 80% methanol solution containing 4 ppm of 2-chloro-L-phenylalanine and filtered through a 0.22 µm membrane. The filtered solution was transferred to a vial for LC-MS/MS analysis.

### 2.9. Statistical Analysis

The analysis of the data was performed using the GraphPad Prism v.8 and R software (v.4.3.1). The experimental values were represented as the mean ± standard error of the mean (SEM). One-way analysis of variance was used to compare the differences between the groups. Statistical significance, strong significance, and very strong significance were considered for *p* < 0.05, *p* < 0.01, and *p* < 0.001, respectively.

## 3. Results

### 3.1. Lp082 Alleviates Depressive-like Behaviors

This study used the CUMS model in mice to evaluate the antidepressant effect of Lp082 through a series of behavioral tests and compared this to the Con and Mod groups (Figure 1A). The results showed that the Mod group exhibited significant depressive-like behaviors, consistent with previous findings [21]. Specifically, compared to the Con group, the mice in the Mod group had significantly reduced activity times in the central area during the OFT (*p* < 0.05) (Figure 1B,C), demonstrated reduced sucrose preferences (*p* < 0.05) (Figure 1D), and showed increased immobility in both the TST and FST (*p* < 0.05) (Figure 1F,G), indicating reduced exploratory behavior, anhedonia, and despair. These results confirmed the success of the CUMS model. Compared to the Mod group, the mice treated with Lp082 showed significant improvements across all four tests (*p* < 0.05), including an increased central area activity time, higher sucrose preference, and decreased immobility time, suggesting a potent antidepressant effect. In addition, the activity time of mice in the open arm can reflect their anxiety to a certain extent. In the present study, there was no significant difference in the open arm activity time between the Con, Mod, and Lp082 groups (Figure 1E). Collectively, these findings indicate that Lp082 significantly improved the depression in the mice.

### 3.2. Lp082 Modulates Neurotransmitter and Inflammatory Cytokine Levels in Serum and Colon

The development of depression is closely linked to neurotransmitter imbalances [22] and inflammatory responses [23]. To examine the effects of Lp082 on the neurotransmitter and inflammatory cytokine levels in the CUMS model mice, we analyzed serum and colon samples from the three groups. The Mod group exhibited significantly lower levels of 5-HT, DA, GABA, and NE in the serum (*p* < 0.05) (Figure 2A,C,D,F) compared to the Con and Lp082 groups, confirming the neurotransmitter imbalances associated with depression [24]. Additionally, there was no difference in the serum LPS levels between the Mod and Con groups (Figure 2J). In contrast, Lp082-treated mice displayed significantly restored levels of GABA, 5-HT, and NE (*p* < 0.05) (Figure 2A,D,F), and their LPS levels were significantly reduced (*p* < 0.05) (Figure 2J).

In the colon, compared to the Con group, the Mod group exhibited significantly lower levels of 5-HT, DA, and GABA (*p* < 0.05) (Figure 2K,M,N) and significantly elevated levels of TNF-α (*p* < 0.05) (Figure 2S) among all groups, indicating an imbalance in neurotransmitter levels and an inflammatory response in the gut. In contrast, Lp082 treatment significantly improved these parameters, with a marked increase in 5-HT (*p* < 0.05) (Figure 2K), the restoration of the GABA levels (*p* < 0.01) (Figure 2N), and significant reductions in the IL-6 and TNF-α levels (*p* < 0.05) (Figure 2R,S). However, the DA levels did not show significant changes (Figure 2M). These results suggest that Lp082 may exert its antidepressant effects by modulating the levels of neurotransmitters and inflammatory cytokines.

### 3.3. Lp082 Affects Neurotransmitter Levels, Inflammatory Cytokines, and Hippocampal Neuronal Markers in the Brain

To further explore the effects of Lp082 on the brain, we measured the neurotransmitter and inflammatory cytokine levels in the brain tissue and assessed hippocampal neuronal markers using Nissl and immunofluorescence staining. Compared to the Con group, significant changes in neurotransmitters were observed in the brains of the mice in the Mod group, as evidenced by the significant downregulation of the 5-HT, DA, and NE levels (*p* < 0.05) (Figure 3A,C,F), which was consistent with the observed decrease in 5-HT levels in depression [25]. Additionally, the GABA levels were also reduced in the Mod group, albeit without statistical significance. Treatment with Lp082 led to a trend of increased GABA levels (Figure 3D). Moreover, Lp082 significantly reduced the LPS levels in the brain (Figure 3I), suggesting its potential anti-inflammatory effects.

The immunofluorescence analysis of the hippocampal region (CA1, CA3, and DG) revealed significant alterations in neuronal and synaptic markers due to CUMS. Compared with the Con group, in the Mod group, the BDNF levels were significantly reduced in the CA1, CA3, and DG regions (*p* < 0.05) (Appendix A), and the PSD-95 levels were significantly lower in the CA3 and DG regions (*p* < 0.05) (Appendix A), indicating impairments in neuroplasticity and synaptic function. In contrast, the Lp082 treatment restored the expression of these markers, suggesting neuroprotective effects. Additionally, the GFAP expression in the DG region was significantly elevated in the Mod group, indicating a neuroinflammatory response. Lp082 treatment significantly reduced the GFAP levels elevated by CUMS (*p* < 0.05) (Figure 3O), further supporting its anti-inflammatory effects. Overall, Lp082 alleviated the depressive-like behaviors of the mice by modulating the neurotransmitter balance, reducing inflammation, and restoring hippocampal neuronal function.

### 3.4. Lp082 Reshapes the Gut Microbiota’s Composition and Function

CUMS exposure significantly altered the composition of the gut microbiota (Figure 4A). Compared to the Con group, several species were significantly increased in the Mod group, including *Alistipes* spp., *Lachnospiraceae bacterium*, *Bacteroides* sp. L10_4, *Muribaculum intestinale*, and *Duncaniella freteri*. Conversely, *B. pseudolongum*, *Faecalibaculum rodentium,* and *Bacteroidales bacterium* were significantly reduced in the Mod group, indicating gut dysbiosis induced by CUMS. These changes were significantly reversed after Lp082 treatment. Species that were reduced in the Mod group, such as *B. pseudolongum*, *F. rodentium*, and *B. bacterium*, increased significantly in the Lp082 group, and species elevated in the Mod group, such as *D. freteri* and *B.* sp. L10_4, were significantly reduced following Lp082 treatment. The probiotic strain used in this study (Lp082) was identified in the Lp082 group, confirming its successful colonization. Additionally, the abundance of a beneficial bacterium, *Akkermansia muciniphila*, was significantly increased in the Lp082 group compared to the Mod group, further supporting the beneficial effects of Lp082 on the gut microbiota composition.

Next, a correlation analysis was performed between *L. plantarum* and other differential bacterial species (Figure 4E). Lp082 was positively correlated with *B. pseudolongum*, *A. muciniphila*, *F. rodentium*, and *M. intestinale* (red), whereas it was negatively correlated with *D. freteri* and *B.* sp. L10_4 (blue).

Finally, the impact of Lp082 on microbial metabolic pathways was examined (Figure 4F). The Mod group exhibited significant changes in metabolic pathways related to neurotransmitter synthesis, inflammation, and energy metabolism compared to the Con group. Notably, pathways involved in L-histidine and L-phenylalanine biosynthesis were downregulated, and pathways associated with L-Glu and L-glutamine biosynthesis were upregulated (*p* < 0.05). Amino acid metabolism is important for the nervous system. For example, phenylalanine is a precursor for DA and NE synthesis [26]. In individuals with depression, dysregulated amino acid levels may contribute to depressive behaviors [27]. In this study, the Mod group mice exhibited depressive-like behaviors. These behaviors were potentially associated with alterations in gut-microbial amino acid metabolic pathways. The Lp082 intervention significantly restored the abundance of several gut-microbial metabolic pathways, including L-phenylalanine biosynthesis. This intervention also led to the marked alleviation of the depression-like behaviors in the mice. Therefore, Lp082 may alleviate depression by restoring the gut microbiota’s structure and function, re-establishing the metabolic balance, and regulating amino acid metabolism.

### 3.5. Lp082 Regulates Serum Metabolite Levels in CUMS Model Mice

In order to investigate which metabolites Lp082 regulates to alleviate depression, a serum metabolomics analysis was performed. The volcano plot displayed the number of significantly upregulated and downregulated metabolites (Figure 5A,B), and the heatmap illustrated the distribution of differential metabolites across the Con, Mod, and Lp082 groups (Figure 5C). Compared with the Con group, the levels of 5-hydroxyindoleacetic acid (5-HIAA) and L-Phenylalanine in the Mod group were significantly increased (*p* < 0.05). After Lp082 treatment, the levels of 5-HIAA and L-phenylalanine were significantly reduced, suggesting that Lp082 may regulate 5-HT and phenylalanine metabolism, contributing to the alleviation of depression.

The serum metabolite enrichment analysis revealed significant alterations in metabolic pathways in the Mod group, particularly in tryptophan metabolism, histidine metabolism, and the biosynthesis of unsaturated fatty acids (Figure 6A). These findings indicate that depression may involve broad metabolic disturbances, including in fatty acid and amino acid metabolism. Compared to the Mod group, Lp082 treatment significantly regulated fatty acid metabolism and positively influenced several other metabolic pathways, including amino acid metabolism, sulfur metabolism, and phospholipid metabolism (Figure 6B). These findings suggest that Lp082 may improve metabolic disorders associated with depression by regulating lipid and amino acid metabolic pathways.

### 3.6. Potential Mechanisms Underlying the Antidepressant Effects of Lp082

A comprehensive analysis was conducted to identify potential mechanisms underlying the observed effects. The Spearman correlation analysis revealed significant relationships between gut microbiota species, metabolic pathways, and biochemical markers (Figure 7). *B. pseudolongum*, *F. rodentium*, and *M. intestinale* were positively correlated with *L. plantarum*, with their abundance significantly increasing in the Lp082 group (*p* < 0.05) (Figure 3D,E). In contrast, *Ligilactobacillus murinus* and *Lachnospiraceae bacterium* showed negative correlations. The increased abundance of metabolic pathways related to L-phenylalanine, L-tyrosine, and L-histidine biosynthesis was positively correlated with DA in the serum and NE, 5-HT, and DA in the brain. The increased production of these amino acids likely contributes to the antidepressant effects of Lp082 by enhancing the synthesis of NE and DA. These findings suggest that Lp082 alleviates depression by modulating the gut microbiota and metabolic pathways (Figure 8).

## 4. Discussion

Probiotics, as an essential component of functional foods, play an indispensable role in promoting human health through their intrinsic properties and fermentation-derived metabolites. In recent years, with advances in gut–brain axis research, probiotics have emerged as a promising dietary intervention strategy to alleviate depression. For instance, the administration of *Lactobacillus rhamnosus* KY16 has been shown to ameliorate depressive symptoms by enhancing the intestinal secretion of 5-hydroxytryptophan (5-HTP) [28]. Furthermore, wheat germ fermented by *Lactiplantibacillus plantarum* has demonstrated significant efficacy in mitigating depression-like behaviors in rodent models [29]. In the present study, the probiotic Lp082, as a potential functional food ingredient, significantly improved depression-like behavior in CUMS mice by restoring the neurotransmitter balance, reducing inflammation, and modulating the gut microbiota. These findings suggest that dietary supplementation with Lp082 may exert antidepressant effects through the modulation of the GBA, highlighting its promise as a microbiota-targeted nutritional intervention for mood regulation.

The disruption of neurotransmitter levels is a key pathophysiological feature of depression, particularly deficiencies in essential neurotransmitters such as 5-HT, NE, GABA, and DA [30,31,32]. The monoamine hypothesis posits that reduced levels of 5-HT and NE are significant contributors to depressive symptoms, which are typically anxiety and low mood [24]. Accumulating evidence underscores the role of imbalances in other neurotransmitters in the development of depression, including GABA and DA. Specifically, the hypothalamic–pituitary–adrenal (HPA) axis, a critical component of the neuroendocrine system, regulates stress responses and mood [33]. Decreased GABA levels impair inhibitory neurotransmission, leading to the excessive activation of the HPA axis, which exacerbates stress responses [34]. Furthermore, decreased DA levels impair the reward system, contributing to intensified depressive symptoms [35].

In this study, the CUMS model was used to examine these pathophysiological features. The results showed that, compared with the Con group, the levels of 5-HT and DA in the sera, colons, or brains of the CUMS mice in the Mod group were significantly decreased and they showed anxiety and depressive behaviors (Figure 2A,C,K,M; Figure 3A,C). Following Lp082 treatment, these neurotransmitters were significantly restored across multiple systems (serum and brain), suggesting that Lp082 may alleviate depressive symptoms by modulating the neurotransmitter balance. Notably, 5-HT is synthesized from tryptophan, with tryptophan hydroxylase converting tryptophan into 5-HTP; subsequently, 5-HTP is converted to 5-HT by tryptophan decarboxylase [36]. More than 90% of the endogenous 5-HT is produced in the gut [37,38]. Moreover, 5-HT can influence the CNS and behavior via the vagus nerve [39]. Additionally, gut-derived 5-HTP has been shown to promote brain 5-HT synthesis, contributing to antidepressant effects [40]. Interestingly, in the colon, significantly higher levels of 5-HT were observed. This suggests that Lp082 may regulate tryptophan metabolism by modulating the gut microbiota, thereby providing the precursor 5-HTP for CNS 5-HT synthesis through the GBA.

Furthermore, the serum metabolomic analysis supported these findings, revealing disturbances in tryptophan metabolism, including L-kynurenine and 3-methylindole, in the Mod group. Lp082 treatment normalized tryptophan metabolism in the CUMS mice. Additionally, Lp082 significantly enhanced the abundance of the superpathway of L-phenylalanine biosynthesis in the gut, which may promote phenylalanine production and provide additional precursors for NE and DA synthesis. The restoration of GABA levels may alleviate stress by inhibiting excessive HPA axis activation, while increased DA levels help to restore reward system function and improve the negative emotional states associated with depression.

The gut microbiota is a crucial component of the GBA and can influence the CNS through the enteric nervous system, peripheral systems, vagus nerve, inflammatory cytokines, and the HPA axis. The gut microbiota plays a critical role in either exacerbating or alleviating depression. On the one hand, it regulates the CNS through amino acid metabolism and neurotransmitter synthesis. On the other hand, gut-derived inflammatory cytokines significantly impact emotional regulation and cognition. Neuroinflammation is considered a key mechanism underlying depression [41], with peripheral pro-inflammatory cytokines activating microglial cells and inducing neuroinflammation and mood disorders [42]. Dysbiosis of the gut microbiota increases intestinal permeability, leading to inflammation [43]. The excessive production of LPS by the gut microbiota triggers the production of inflammatory cytokines such as IL-6, IL-1β, and TNF-α [44,45], further disrupting the HPA axis, increasing brain inflammation, and promoting neurodegeneration, which impairs hippocampal neurogenesis [46]. When hippocampal neurons are damaged, proteins such as BDNF, PSD-95, and GFAP are disrupted. BDNF is a key mediator of synaptic plasticity, promoting neuronal growth and synapse formation [47]. PSD-95 is a scaffolding protein central to synaptic plasticity [48], whereas GFAP is a marker of astrocyte activation [49].

In this study, CUMS-induced dysbiosis in the Mod group led to significant increases in TNF-α in the gut. These inflammatory cytokines likely induced neuroinflammation through the vagus nerve or HPA axis, resulting in reduced neuroplasticity and impaired hippocampal neurogenesis. The Lp082 intervention reshaped the gut microbiota in the CUMS mice, significantly increasing the abundance of beneficial microorganisms such as *B. pseudolongum*, *F. rodentium*, and *A. muciniphila* [50,51,52]. It also reduced the levels of inflammatory cytokines in the gut, blood, and brain. Furthermore, Lp082 significantly increased the BDNF and PSD-95 levels in the hippocampi of the CUMS mice and decreased their elevated GFAP levels. These results suggest that Lp082 may exert antidepressant effects by modulating the gut microbiota balance, alleviating the gut and neuroinflammatory responses, and reducing neurodamage.

## 5. Conclusions

In conclusion, this study suggests that Lp082, as a probiotic with potential applications in functional foods, may alleviate CUMS-induced depression in mice by modulating the composition and function of the gut microbiota. Lp082 appears to influence amino acid metabolic pathways in the gut, including those related to L-phenylalanine and L-tyrosine biosynthesis. These pathways facilitate the synthesis of key neurotransmitters, such as NE, 5-HT, and DA, which may subsequently impact brain function via the enteric nervous system or the HPA axis, thereby reducing depressive symptoms. Additionally, Lp082 increases the abundance of beneficial gut bacteria, such as *B. pseudolongum* and *A. muciniphila*, which, in turn, may reduce gut inflammation and consequently neuroinflammation. This reduction in inflammation likely enhances neuroplasticity and promotes neurogenesis in the hippocampus. These findings highlight the potential of Lp082 as a functional food ingredient to improve mental health through GBA modulation.

## Figures and Tables

**Figure 1 foods-14-00813-f001:**
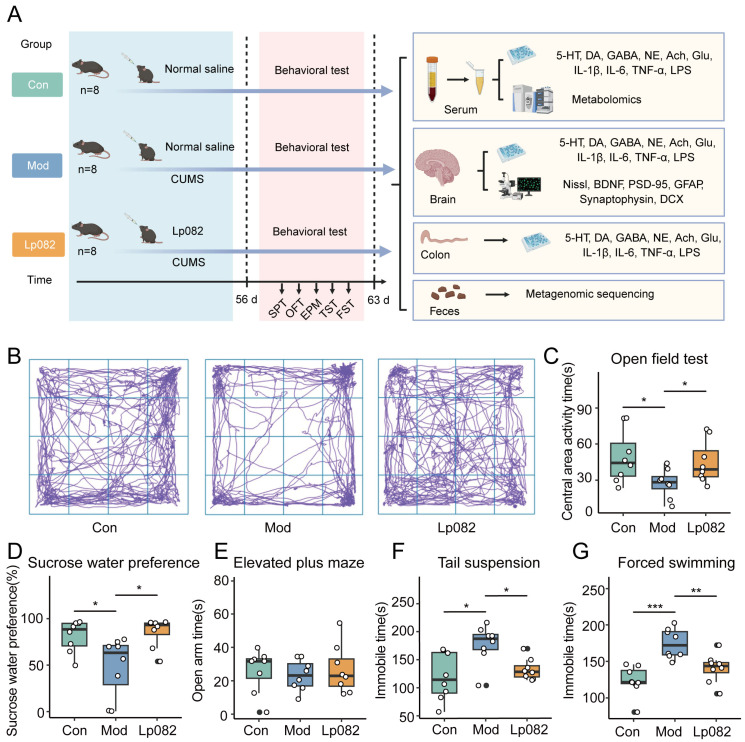
Lp082 alleviates CUMS-induced depressive-like behavior. (**A**) Flow chart of the experiment. (**B**) Trajectory diagram of mice’s activity in the OFT. (**C**) Activity time of mice in the central area in the OFT. (**D**) Sucrose preference rate of mice in the SPT. (**E**) Activity time of mice in the open arm in the EPM. (**F**) Mouse immobility time in the TST. (**G**) Immobility time of mice in the FST. Data are expressed as the mean ± standard error of the mean (SEM). * *p* < 0.05, ** *p* < 0.01, *** *p* < 0.001.

**Figure 2 foods-14-00813-f002:**
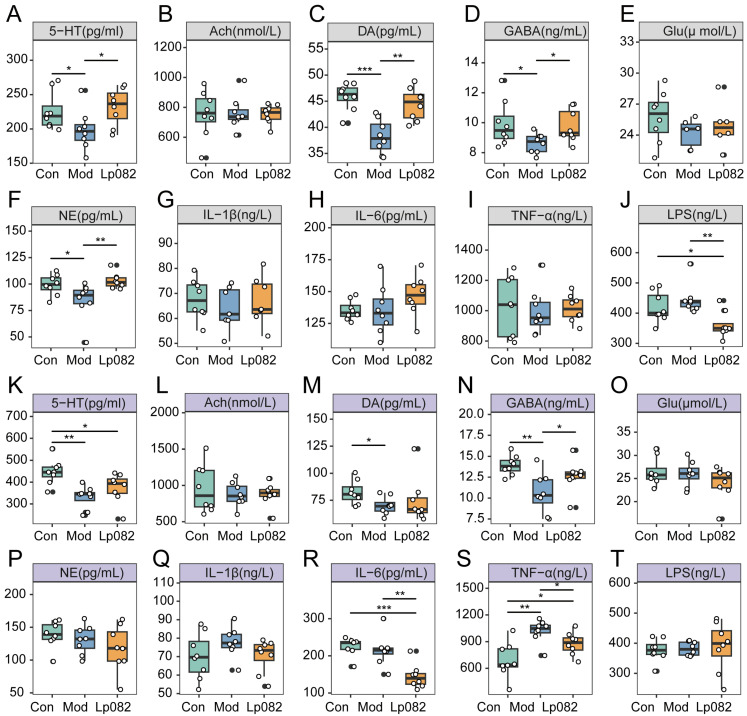
Lp082 regulates neurotransmitter and inflammatory cytokine levels in serum and colon. (**A**–**J**) Neurotransmitter and inflammatory cytokine levels in serum. (**A**) Serotonin (5-HT) levels, (**B**) acetylcholine (Ach) levels, (**C**) dopamine (DA) levels, (**D**) gamma-aminobutyric acid (GABA) levels, (**E**) glutamate (Glu) levels, (**F**) norepinephrine (NE) levels, (**G**) interleukin-1β (IL-1β) levels, (**H**) IL-6 levels, (**I**) tumor necrosis factor-alpha (TNF-α) levels, (**J**) lipopolysaccharide (LPS) levels. (**K**–**T**) Neurotransmitter and inflammatory cytokine levels in the colon. (K) 5-HT levels, (**L**) Ach levels, (**M**) DA levels, (**N**) GABA levels, (**O**) Glu levels, (**P**) NE levels, (**Q**) IL-1β levels, (**R**) IL-6 levels, (**S**) TNF-α levels, (**T**) LPS levels. Data are expressed as mean ± SEM. * *p* < 0.05, ** *p* < 0.01, *** *p* < 0.001.

**Figure 3 foods-14-00813-f003:**
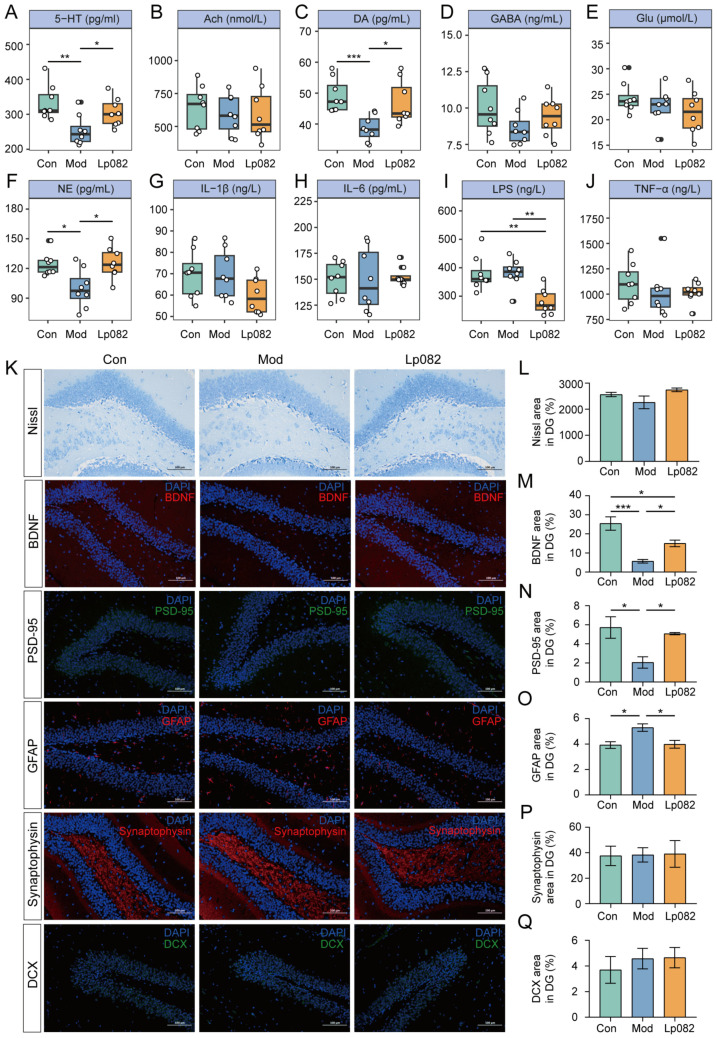
Effects of Lp082 on neurotransmitter and inflammatory cytokine levels in brain tissue and neuron-related markers in the DG region of the hippocampus in CUMS model mice. (**A**–**J**) Neurotransmitter and inflammatory cytokine levels in brain tissue: 5-HT (**A**), Ach (**B**), DA (**C**), GABA (**D**), Glu (**E**), NE (**F**), IL-1β (**G**), IL-6 (**H**), LPS (**I**), and TNF-α (**J**). (**K**) Nissl staining was used to observe the condition of neurons in the hippocampal DG region, and immunofluorescence staining showed the expression of brain-derived neurotrophic factor (BDNF), postsynaptic density protein 95 (PSD-95), glial fibrillary acidic protein (GFAP), synaptophysin, and doublecortin (DCX). (**L**–**Q**) Percentage of positive expression of the above markers in the DG area of the hippocampus: Nissl staining (**L**), BDNF (**M**), PSD-95 (**N**), GFAP (**O**), synaptophysin (**P**), and DCX (**Q**). Data are expressed as mean ± SEM. * *p* < 0.05, ** *p* < 0.01, *** *p* < 0.001.

**Figure 4 foods-14-00813-f004:**
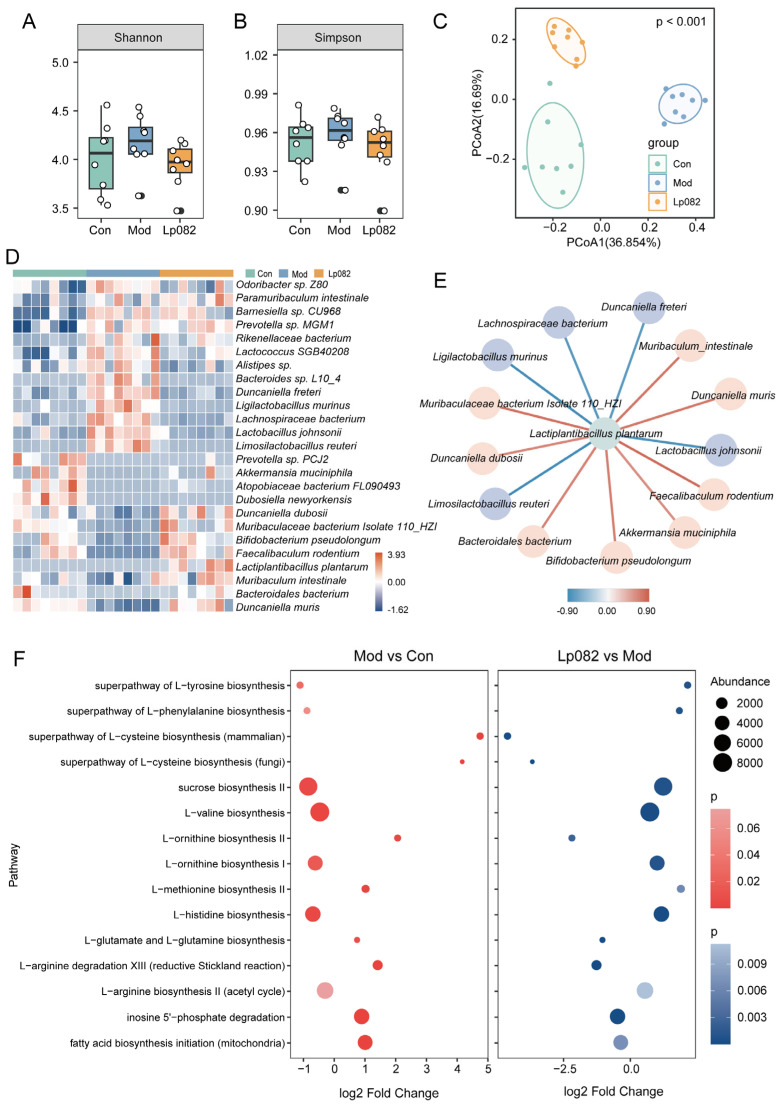
Lp082 ameliorated the gut microbiota imbalance. (**A**) Shannon index. (**B**) Simpson index. (**C**) Principal coordinate analysis (PCoA) of microbial communities. (**D**) Heatmap of differential microbial abundance at species level. (**E**) Correlation network between Lp082 and different species. (**F**) Microbial metabolic pathways.

**Figure 5 foods-14-00813-f005:**
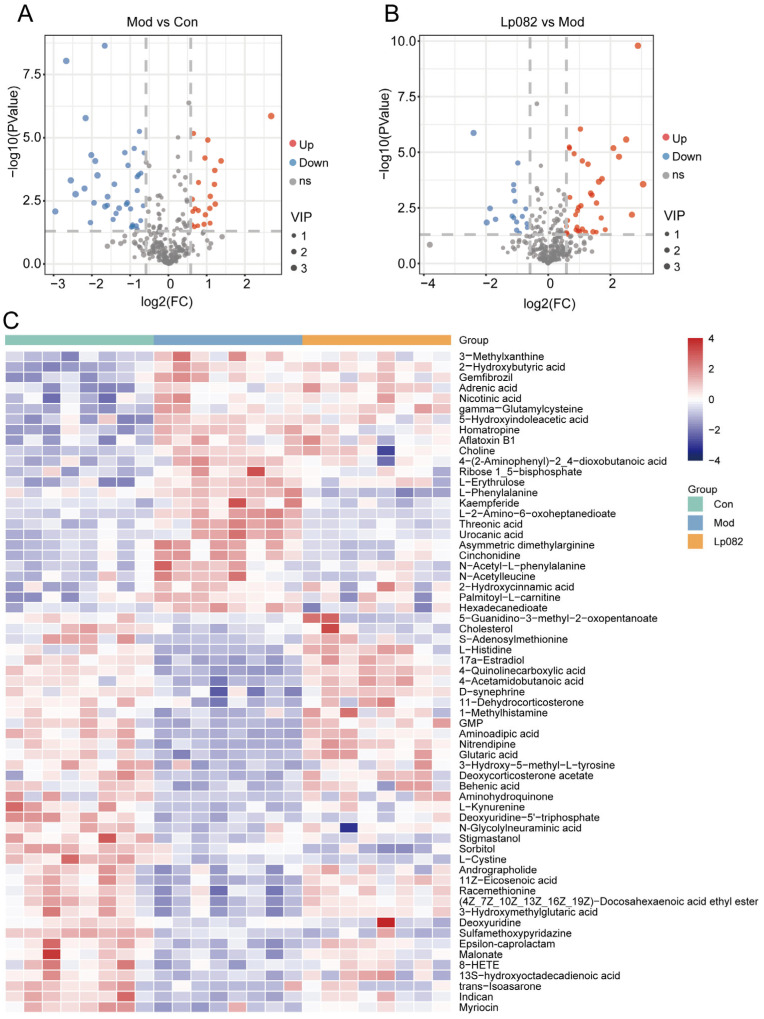
Lp082 regulates serum metabolite levels in CUMS model mice. (**A**,**B**) Metabolite volcano plot. (**C**) Heatmap of differential metabolites.

**Figure 6 foods-14-00813-f006:**
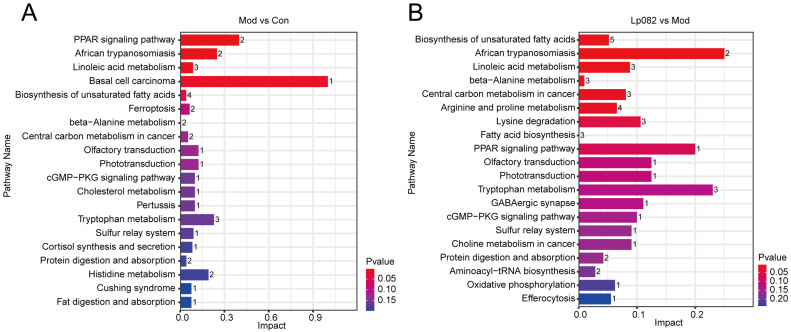
Enrichment map of differential metabolites. (**A**,**B**) Differential metabolite enrichment bar diagram.

**Figure 7 foods-14-00813-f007:**
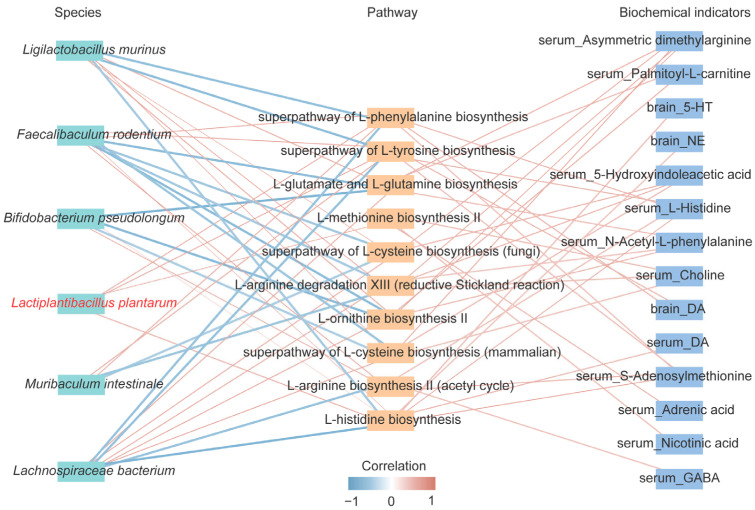
Correlation analysis. Correlation network map of gut microbiota, metabolic pathways, and biochemical indicators. The red line represents a positive correlation, whereas the blue line denotes a negative correlation.

**Figure 8 foods-14-00813-f008:**
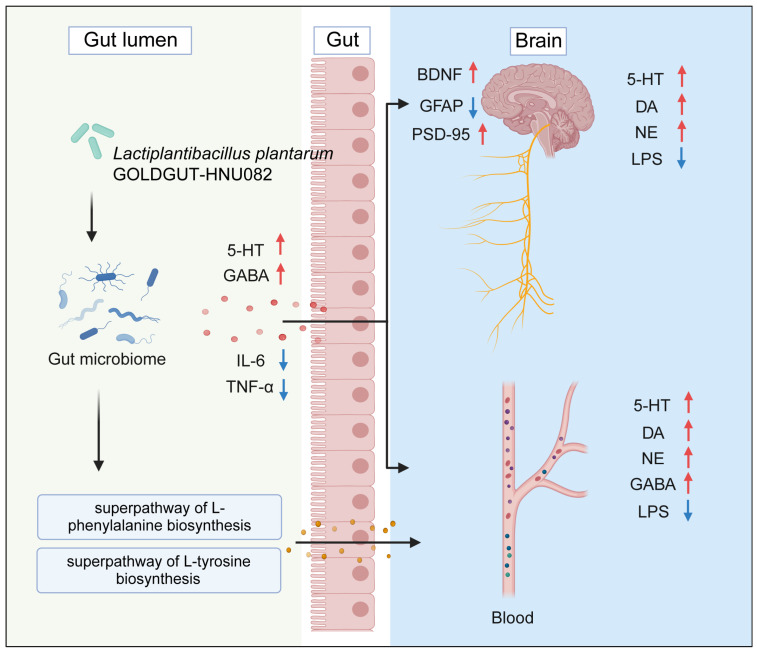
Potential mechanisms by which Lp082 ameliorates depression and neurological damage in CUMS model mice.

## Data Availability

The datasets generated during the current study are available in the National Center for Biotechnology Information (NCBI) repository (PRJNA1193219).

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
