# Peer review of "Lactiplantibacillus plantarum GOLDGUT-HNU082 Alleviates CUMS-Induced Depressive-like Behaviors in Mice by Modulating the Gut Microbiota and Neurotransmitter Levels"

_foods, 2025, doi:10.3390/foods14050813_

Round 1
Reviewer 1 Report
Comments and Suggestions for Authors
2/10/2025
Lactiplantibacillus plantarum GOLDGUT-HNU082 alleviates CUMS-induced depressive-like behaviors in mice by modulating the gut microbiota and neurotransmitter levels
The intestinal microbiota plays a crucial role in the synthesis and release of various hormones and neurotransmitters associated with gut-brain axis, which subsequently modulate the brain function and host behaviour. Psycho-biotics, a subset of probiotics, are emerging as promising agents for enhancing mental health through the modulation of the gut microbiota. Reports suggest that probiotics can improve severe depression and intestinal dysfunction in humans. These studies found a close relationship between the gut microbiome and the pathogenesis of depression.
This study examines the effects of Lactiplantibacillus plantarum (Lp082) on chronic unpredictable mild stress (CUMS)-induced depression in mice. The findings suggested that Lp082 modulated the composition of the gut microbiota, alleviated depressive-like behaviours, restored the balance of vital neurotransmitters, reduced levels of inflammatory cytokines and enhanced brain neuroplasticity in CUMS mice.
The article is well-written and comprehensive, with a robust study design that significantly enhances our understanding of how probiotic modulation of the intestinal microbiota affects brain function. However, the article needs transparency/ clarity in data presentation and logical interpretation of findings. The data presentation and interpretation could benefit from greater transparency and clarity. The interpretation of the results is somewhat misleading and overly generalized. Some results, such as those that were not statistically significant but were reported as significantly different, need to be reassessed, and the corresponding statements should be revised to avoid confusion and to improve the article.
Other Comments:
Line 92: “Lp082 was isolated from Yucha, a traditional fermented food of the Li ethnic group 92 in Hainan Province, China.” Could you please provide how it was isolated? and characterized? Was the isolation done in this study or previously been done? needs more detail
Line 100: “Mice were housed in SPF-grade animal facilities” Please expand the term SPF
Line 102: “three groups: Con (n = 8), Mod (n = 8)” Please expand the term ‘Con’ and ‘Mod’ at first appearance.
Line 119: “field test, the box must be disinfected with 75% medical alcohol to eliminate odors.” Rephrase the sentence and structure it in past tense. What is meant by medical alcohol? Is it Ethanol? Please mention clearly and precisely.
Line 157-165: Could you please provide details about the supplier/ manufacturer’s information for the reagents/ antibodies used in the study such as neurrotrophic factor (BDNF), doublecortin (DCX), synaptophysin, postsynaptic density protein 95 (PSD-95), and glial fibrillary acidic protein (GFAP), conjugated with red or green fluorescent labels, fluorescently labeled secondary antibodies, stained with DAPI, antifade reagent.
Line 205: Could you please shed some light on “Figure E”
Line 217-218: “The Mod group exhibited significantly lower levels of NE, GABA, 5-HT, and DA in serum (P < 0.05) (Figure 2A, C, D, and F)” Could you correct the order according to the figure number.
Line 220-221: “Additionally, the level of LPS in serum showed an increasing trend in the Mod group (Figure 2J).” The statement is unclear as the trend is not evident to be increasing.
Line 232-233: “Lp082 may exert its antidepressant effects by modulating neurotransmitter and inflammatory cytokines levels in both serum and colon tissues.” IL-6, TN-alpha levels differ in serum so contradicts the statement. Could you explain the levels of pro-inflammatory cytokines differences in serum vs colon? Such as for levels of IL-6, TN-α, IL-1β, were they significantly different? Why is there an opposite trend for IL-6 in serum and colon tissues?
Line 249: “consistent with neurotransmitter imbalances observed in depression.” The statement is unclear, clarify / please provide reference to it.
Line 256-257: “CA1, CA3, and DG region (P < 0.05) (Figure S1C† and S2C†),” The values do not follow the figures, please provide correct Figure numbers.
Line 257-258: “PSD-95 levels were signifi-257 cantly lower in the CA3 and DG region (P < 0.05) (Figure S1D† and S2D†),” Please provide correct Figure numbers.
Line 380-382: “The results showed that compared with the Con group, the levels of NE, 5-HT, GABA and DA in the serum, colon or brain of CUMS mice in the Mod group were significantly decreased” The statement is misleading. Please revisit the data and correct the statement such as NE levels were not significantly decreased in colon, similarly GABA levels were not significantly reduced in brain.
Also please mention the figure number the statements are referring to.
Line 382-383: “Following Lp082 treatment, these neurotransmitters were significantly restored across multiple systems (gut, serum, and brain)” The statement is misleading. Please revise and correct the statement as the results were not significantly different for all three i.e. gut, serum, and brain.
Line 422-423: “In this study, CUMS-induced dysbiosis in the Mod group led to significant increases 422 in LPS, TNF-α, and other inflammatory cytokines in the gut and blood.” The statement is misleading. Please revise and correct the statement.
Please check the formatting and correct where applicable such as
Line 28: “ like Bifidobacterium and improving metabolic…” italicise Bifidobacterium.
Line 69: “cannabinoid system.1” could be reformatted to “cannabinoid system [1].”
Line 72: “depressive-like behaviors in mice.[8]” could be reformatted to “depressive-like behaviors in mice [8].”
Line 225:” (Figure 2K M and N),” could be reformatted to “(Figure 2K, M and N),”
Line 248: “DA and NE levels. (P < 0.05)” could be reformatted to “DA and NE levels (P < 0.05)”

Reviewer 2 Report
Comments and Suggestions for Authors
Comments in attached file

Author Response
Reviewer #2:
This study suggests that Lp082 may alleviate anxiety- and depression like behaviors induced by CUMS in mice by modulating the composition and function of the gut microbiota. Some specific comments to your work:
Response: Thank you for taking time to read and consider our manuscript. According to your suggestions, we have carefully revised manuscript. Your valuable suggestions and will be of great help to us.
Comments 1: The authors could add this reference in the introduction:
Joanna Godzien, Bartlomiej Kalaska, Leszek Rudzki, Cecilia Barbas-Bernardos, Justyna Swieton, Angeles Lopez-Gonzalvez, Lucyna Ostrowska, Agata Szulc, Napoleon Waszkiewicz, Michal Ciborowski, Antonia García, Adam Kretowski, Coral Barbas, Dariusz Pawlak, Probiotic Lactobacillus plantarum 299v supplementation in patients with major depression in a double-blind, randomized, placebo-controlled trial: A metabolomics study. Journal of Affective Disorders 2025, 368, 180-190, doi.org/10.1016/j.jad.2024.09.058.
Response 1: Thank you for your valuable advice. Relevant literature has been cited. Please lines 59-62.
Comments 2: It would have been interesting to have a larger number of animals per group, 8 is not enough.
Response 2: Thank you for your valuable feedback. During the experimental design phase, we conducted an extensive review of numerous studies and found that the majority of animal experiments involving mice utilized a sample size of at least 6 animals per group (n≥6). To enhance the reliability of our experimental results, we decided to include 8 mice in each group.
Comments 3: The authors should indicate whether the dose was daily and when it was given.
“During this period, the Con and Mod groups were administered 200 μL of physiological saline via gavage, while the Lp082 group received 200 μL of Lp082 suspension (1×10⁹ CFU) via the same metho”.
Response 3: Thank you for pointing out this issue. The specific timing and frequency of administration have already been described in the manuscript. Please refer to lines 114-117 for details.
Comments 4: Why the trial time was set at 8 weeks?
Response 4: In the experimental design phase, we conducted an extensive review of the literature and found that the Chronic Unpredictable Mild Stress (CUMS) protocol, commonly used to establish a mouse model of depression, typically spans a duration of 6 to 8 weeks. Therefore, we adopted an 8-week CUMS protocol to construct the mouse model of depression.
References: Chevalier, G.; Siopi, E.; Guenin-Macé, L.; Pascal, M.; Laval, T.; Rifflet, A.; Boneca, I.G.; Demangel, C.; Colsch, B.; Pruvost, A.; et al. Effect of Gut Microbiota on Depressive-like Behaviors in Mice Is Mediated by the Endocannabinoid System. Nat Commun 2020, 11, 6363, doi:10.1038/s41467-020-19931-2.